# Willingness to adopt green house gas mitigation measures: Agricultural land managers in the United Kingdom

Asma Jebari[1], Zainab Oyetunde-Usman[2]*, Graham A. McAuliffe[1], Charlotte-Anne Chivers[3], Adrian L. Collins[1]

1 Net Zero and Resilient Farming, Rothamsted Research, North Wyke, Okehampton, Devon, United Kingdom, 2 Net Zero & Resilient Farming, Rothamsted Research, West Common, Harpenden, United Kingdom, 3 Countryside and Community Research Institute, University of Gloucestershire, Cheltenham, United Kingdom

* zainab.oyetunde-usman@rothamsted.ac.uk, zainabus23@gmail.com

## Abstract

Rapid uptake of greenhouse gas (GHG) mitigation measures is central to reducing agricultural and land use emissions and meeting the UK Net Zero policy. The socioeconomic challenges and barriers to uptake are poorly understood, with yet unclear structural pathways to the uptake of GHG mitigation measures. Using an online survey of 201 agricultural land managers across the UK, and applying multiple linear regression and stepwise regression analysis, this research established farm and farmers' factors influencing perceptions and willingness to adopt GHG mitigation measures. The results consistently show that farm sector, farmers' business perception, and labour availability influence willingness to adopt GHG mitigation measures. Based on the farmers' qualitative feedback, other barriers to adoption include costs and concerns for profitability, lack of flexibility in land tenancy contracts, poor awareness and knowledge of the application of some GHG mitigation measures, perception about market demand e.g bioenergy crops, and scepticism about the future impacts of adopting varying GHG mitigation measures. In the midst of the ongoing net zero transition, this study identifies existing barriers to the uptake of GHG mitigation measures, and specifically, a substantial gap between farmers and the science of GHG mitigation measures and the need to incentivise a farm and farming community-led policy interventions to promote adoption of GHG mitigation measures.

## 1.0 Introduction

Agriculture in the UK accounts for $\sim 71\%$ of land use, of which grassland and crop production constitute $\sim 72\%$ and 26%, respectively [1]. The agricultural sector accounts for 11% of total greenhouse gas (GHG) emissions wherein $\sim 58\%$ of these GHG are generated by livestock in the form of $CH_4$ from enteric fermentation (eructation primarily) and manure management (i.e., storage and subsequent application as organic fertilisers). The varied impacts of GHG emissions in the UK include increasing incidences of drought associated with changing

**Data Availability Statement:** https://doi.org/10.23637/rothamsted.99058.

**Funding:** The first Author - AJ, was financially supported by the Science Initiative Catalyst Award

(SICA) programme, a Rothamsted Research internal UKRI (UK Research and Innovation) Biotechnology and Biological Sciences Research Council (UKRI-BBSRC) award. GM and ALC acknowledge funding from Rothamsted Research's Institutional Strategic Programmes 'Soil to Nutrition' (S2N) supported by UKRI-BBSRC BBS/E/C/000I0320 & BBS/E/C/000I0330 and 'Resilient Farming Futures' supported by UKRI-BBSRC BB/X010961/1. There was no additional external funding received for this study.

**Competing interests:** The authors declare no known competing interests that could have appeared to influence the work reported in this paper.

weather and climate [2], distributions of pests and diseases [3], and the occurrence of invasive weeds and non-native species [4], which directly or indirectly impact agricultural productivity, nutrition and fluctuations in agricultural commodity prices [5]. Since agriculture and land use make an important contribution to GHG emissions, improved mitigation in this sector is central to achieving the UK Net Zero policy objectives in 2050 (CCC 2020) [6], and more specifically, in accordance with the UK Committee on Climate Change (CCC), a 64% reduction in GHG emissions from the agriculture and land-use sector (CCC 2020). Policymakers, therefore, face the challenge of developing and implementing effective GHG abatement strategies for agriculture.

Numerous mitigation strategies, both extant and at the prototype stage, are being proposed to tackle climate risk through reducing and/or offsetting emissions and improving production efficiency [7]. Whilst the technically feasible environmental impacts of available and emerging mitigation measures have been reported in existing literature, corresponding implications for energy, food security, and especially farmer attitudes towards the adoption of improved practices have received scant attention [8]. Nonetheless, understanding attitudes, awareness and intentions to change current practices is one of the critical indicators of the uptake of GHG mitigation measures.

Information on the attitudes of agricultural land managers towards different mitigation measures is needed, as agricultural land managers are the ones who ultimately make most of the management decisions for 71% of the UK land area [9]. Understanding how agricultural land managers perceive GHG mitigation measures and how this affects their willingness to adopt them is critical for developing effective climate change response strategies for the agricultural sector. Our study enriches existing empirical evidence [10–12] by undertaking an assessment using an extensive survey comprising key information surrounding mitigation measures (e.g., economic, energy, food security) to address the following objectives: i) understanding farmers' willingness to adopt GHG mitigation measures, and ii) assessing the critical factors for predicting farmers' willingness to adopt GHG mitigation measures. In the next section, we discuss the overview of GHG mitigation measures considered in this study, their importance and prospects for tackling climate risk.

## 2.0 Brief overview of GHG mitigation measures and impacts on emissions reductions and soil organic carbon

Climate change is caused by a range of gases collectively known as GHGs and these are defined as gases that absorb and re-emit heat with a continuous warming effect on the planet's atmosphere [13]. Nitrous oxide ($N_2O$), methane ($CH_4$), and carbon dioxide ($CO_2$) are common emissions from agriculture and land use activities and, respectively, contribute 14.5%, 24.8% and 5.5% to the total UK agricultural emissions [14]. Reducing measures involves maintaining or enhancing existing efficacy for controlling emissions potential. Grasslands are, for example, home to a range of flora and fauna and have the potential to improve biodiversity and reduce GHG emissions due to their high carbon sequestration potential [15]. Certain factors such as land use change and intensification management threaten the multifunctionality of grassland areas. A typical example is the conversion of grassland to cropland for livestock grazing, this usually leads to intensive grazing which affects the depletion of above-ground biomass and the rate and extent of pollination [16]. Also, poor or suboptimal, fertiliser management and livestock performance, on the other hand, can lead to excessive losses of GHG including $N_2O$ and $CH_4$ both of which are powerful compounds thought to be around 27 (biogenic $CH_4$) and 270 times ($N_2O$) more warming than carbon dioxide ($CO_2$), respectively [17–19].

Grazing management practices, such as reducing livestock density and application inputs (e.g., nitrogen fertilisation), have significant impact in reducing GHG emissions by up to 65% [20]. Besides impacting emission reductions, grazing management tools such as rotational grazing and herbal ley practices can significantly increase herbage dry matter production and improve animal daily live-weight gain [21].

Enteric fermentation emissions ($CH_4$) and manure management emissions ($CH_4$ and $N_2O$) from livestock production account for up to 231.3, 32.3 and 18.3 Mg of $CO_2$ eq. $yr^{-1}$ on dairy, sow and pig farms, respectively [22]. Traditional management of slurry and farmyard manure relies on an agronomic approach that is odour-offensive and poses a risk to waterbodies through direct water pollution, emissions of GHG and release of faecal contamination which can endanger human and animal health [23]. Anaerobic digestion technologies provide solutions to traditional slurry management problems (waste and manure management) and are also beneficial for energy generation [24]. For manure management options, anaerobic digestion accelerates the degradation process of manure without negatively impacting the environment [25] and biologically breaks down organic matter into biogas, water and residual matter (digestate) [26]. Biogas is a renewable energy source and its $CH_4$ composition allows its use as fuel [27] and digestate is suitable for use as a fertiliser or soil conditioner [26]. Anaerobic digestion can reduce farm GHG emissions by up to 44% [27]; there are, however, poor incentives for the widespread adoption of manure-based farm-scale anaerobic digestion in the UK [27].

Emission intensity from livestock production contributes to the aggregate agricultural GHG emissions e.g., Methane ($CH_4$) and Ammonia ($NH_3$) - Methane emissions are primarily generated through enteric fermentation of livestock feed such as grazed grass and nutritional produce, and $NH_3$ emissions are generated from urine and faeces [28]. Addressing livestock emissions targets using feeding and breeding practices, can include adapting nutritional strategies that reduce enteric $CH_4$ emissions through the use of feed additive supplementation, such as dietary lipids and nitrates which in the specific case of supplements can reduce $CH_4$ enteric emissions by up to 45% [29]. Feeding supplements when incorrectly adapted can induce a risk of toxicity and impair animal performance.

Soil amendments are crucial to enhancing soil fertility through improving soil organic carbon (SOC) for plant growth and agricultural productivity [30]. Soil amendments (e.g., basalt amendment, biosolids) enhance SOC accumulation by up to 17% in the case of biosolids applications, and 4-fold with basalt rock dust amendment [31,32], and can substantially increase crop yield over what inorganic fertiliser achieves [33]. There are, however, contrasting reported effects arising from organic amendments; for example, the application of varying organic amendments (rapeseed meals, soybean meals and cattle manure) to winter wheat and summer maize shows that soybean meal and cattle manure were more effective at improving soil quality than rapeseed meals [34], Also, in a long-term experiment, organic amendments were found to be slow in affecting yield when applied solely, compared to inorganic fertilisers, but performed excellently when partially combined with inorganic fertilisers [35]. Application of soil amendments, despite their importance, can pose risks to health and the environment; this is because soil amendments contain a range of pollutants and involve the use of microplastics which can have adverse effects on the environment [36].

Planting cover crops and trees are typical offsetting measures to enhance SOC [37]. Cover crops benefit the soil and the environment by reducing nitrogen losses, enhancing water retention, and preventing soil erosion and sediment loss [38], and can also be reimagined as bioenergy crops and serve as sources of renewable natural gas production and carbon storage [39]. Similarly, trees and hedgerows on agricultural land contribute to microclimate amelioration, providing shelter for livestock, and improving the conservation of soil and water [40–45]. In addition, agroforestry is effective in reducing soil erosion by over 50% [46] and can promote

carbon sinks and GHG mitigation across arid and semiarid regions [47]. The effects of agroforestry on carbon sinks can, however, vary based on the types; for example, between woodland and silvopastoral agroforestry systems, measures of above-ground carbon indicated an overall increase in SOC, while the silvopastoral system was predicted to achieve a higher level of carbon storage than equivalent areas of separate woodland and pasture [48]. Although the high initial investment is an identified limiting factor, the economic and environmental benefits of agroforestry are numerous and include product diversification (e.g., food, recreation, leisure, foliage, biochar, etc) [49].

Bioenergy crops provide multiple functions through limiting the emissions of GHG whilst also acting as sources of renewable energy. Bioenergy crops include feedstocks (e.g., oil seeds, wheat, maize) and non-feedstock crops (e.g., Miscanthus, Willow crops) used to produce liquid biofuel and biogas through an anaerobic digestion approach. In the UK, *Miscanthus* is an example of a dedicated energy crop which is non-invasive, with low production costs (low to no addition of fertiliser) for producing energy and improving removal of GHG emissions. Bioenergy is the second largest source of renewable energy in the UK and a viable sustainable alternative to conventional fossil fuel energy sources as it improves energy security [50–52]. The biodiversity impacts of bioenergy crops include habitat, soil and water quality improvement, support for climate regulation [50–53], and improved farm-scale biodiversity on agricultural land [54]. The impact of bioenergy crops on attaining the Net Zero target is well established and it is context-dependent [55]. Non-food bioenergy crops, for example, directly compete with agricultural and ecosystem services and thus emphasise the importance of landscape contexts in scaling bioenergy crops [56]. Although there is some evidence concerning farmers' preferences and contentment with the economic and environmental benefits of bioenergy crops [57], geographically limited markets due to poorly distributed biomass processing stations and high transportation costs represent barriers to the adoption of bioenergy crops in the UK, also, the use of bioenergy crops is associated with increased claims for land use [58,59].

With varying evidence of the impact of GHG mitigation measures, understanding approaches to scaling adoption is critical. This study contributes to such discussions and provides insight into existing gaps. The next section highlights the examples of factors promoting/limiting the adoption of GHG mitigation strategies as described by past studies; this is followed by a description of the data and estimation strategy adopted in this study.

## 3.0 Summary of past studies on adoption of mitigation strategies

Mitigation and adaptation practices are the two main responses to climate change recognised by the United Nations Framework Convention on Climate Change (UNFCCC); essential in reducing expected climate change impact on the environment [60]. Across the Global North, mitigation measures are becoming popular as effective approaches to reaching Net Zero policy strategies [61–63]. Identified constraints across economic, innovation-diffusion, and adopters' perception paradigms to adoption are comprehensively defined [64]. Economic constraints are, for example, one of the prominent limitations; this is because emission reduction practices can have either a positive or negative impact on farmers' income [65]. Also, farmers' willingness to adopt is economically incentivised and transitioning happens only if practices are perceived to be more cost-effective than the current practices, they are adopting [66] For capital-intensive mitigation practices, profitability and long-term pay-off periods are major constraints to adoption [67].

In the adoption literature, complexities in the adoption processes of agricultural innovations are subjective [68], and are defined by adopters' perceptions considering their knowledge, behaviours, beliefs, and attitudes which are intrinsic to motivations to engage in

emission reduction practices [68–72]. These factors share a similar stage in defining adoption and can independently, or jointly, define motives for adoption [71,72]. To highlight this, adoption is, for example, defined by farmers' risk perception, level of risk aversion and the availability and knowledge of strategies to cope with varying risks [72,73]. Also, behaviours and beliefs about mitigation and sustainable agricultural practices are important contributors to farmers' behavioural intention to adopt [74]. Beliefs are complex and vary considerably and according to a study are intuitively dependent on what farmers attribute their perception of climate change and mitigation practices to [75]. The roles of beliefs and behavioural factors are becoming popular and are found to be important in farmers' decision-making [76]. Adopters' perception includes personal and other physical characteristics defined by farm and farmers' characteristics which can directly impact adoption decisions [77–79]. Constraints to adoption can, for example, include competing use of farm resources between practices, high demand for other resources such as labour, lack of adequate knowledge, and farmers' unfamiliarity with technology [77]. Table 1 below highlights the examples of adoption studies on factors limiting/promoting the uptake of GHG mitigation measures.

## 4.0 Data and estimation strategy

### 4.1 Data

This study utilised a cross-sectional online survey to sample 201 farm households across different farm systems in the UK, to understand perceptions for adoption of GHG measures and the corresponding predictive factors. It included a range of question types such as Likert-scale, multiple choice, and free-text. The survey questions spanned different GHG mitigation measures categorised into reduction measures, offsetting measures and bioenergy crop production.

**Table 1. Summary of related studies on adoption factors for GHG mitigation measures.**

| Authors | Country | Mitigations techniques | Examples of adoption factors |
|---|---|---|---|
| Haden et al. 2012 [80] | United States | Mitigation and Adaptation practices | Mitigation is driven by psychologically distant concerns and beliefs about climate change, while adaptation is driven by psychologically proximate concerns for local impacts. |
| Barnes et. al, 2022 [81] | United Kingdom (Scotland) | Climate-smart farming | Past behaviour is a strong predictor of intention to increase on-farm forestry and renewables. |
| Dijk et al, 2015 [82] | Netherlands | Agri-environmental schemes (AES)–Ditch bank management and protection of meadows birds. | Attitude and perceived personal ability to participate in these AES are associated with the intention of farmers to participate in ditch bank management. |
| Mishra et al 2018 [77] | United states | Row cropping | Farmers who grew row crops, had irrigation facilities, and were in favour of crop diversification were significantly more likely than their respective counterparts to adopt more sustainable agriculture practices. In contrast, a lack of adequate knowledge about sustainable farming and an unfamiliarity with technology significantly and negatively related to less adoption of sustainable agriculture practices. |
| Howley 2013 [70] | Ireland | Forest management | Farmers with relatively stronger economic motivations for forest ownership were found to be much more likely to harvest thinnings whereas the opposite was true of those with more lifestyle-orientated objectives. |
| Feliciano et al. 2013 [83] | North East Scotland | Land-use-based mitigation practices | Economic, social, political and institutional factors affect the uptake of mitigation practices in the region. |
| Feliciano et al. 2014 [66] | North East Scotland | Climate change mitigation practices | Barriers to the implementation of mitigation practices are mainly related to physical-environmental constraints, lack of information and education and personal interests and values. Similarly, enablers are also related to physical-environmental factors and personal interests and values. |
| How et al. 2018 [67] | Denmark, Italy and the Netherlands | Manure treatment technologies | Positive: Pressure from government. Negative: Economic factors, lack of investment capital, high processing cost, and long payback period. |

Each of these measures was described in the survey, including their potential impacts on mitigating GHG emissions (see Table 2). The measure of farmers' willingness to adopt GHG mitigation was described based on a 5-point scale as follows: 1 = very unwilling; 2 = quite unwilling; 3 = neutral, 4 = quite willing and 5 = very willing. The survey data was followed up with open and closed questions to understand more underlying details of farmers' adoption decisions. Here, questions were included on farm and farmers' characteristics such as farm sector, farm size, tenure status, number of full-time equivalent workers, farming years of experience, age group, farmers' perception of business performance, motivation to uptake friendly practices, and measures of willingness to adopt GHG reduction measures. The survey design,

**Table 2. Description of GHG mitigation measures (Adapted from Jebari et al. 2024 [8]).**

| GHG measures | Description and potential impact of GHG mitigation measures |
|---|---|
| Decrease N fertiliser use | Decreasing the use of nitrogen fertiliser could result in up to 70.6% reduction in $N_2O$-N emissions of managed grasslands whilst improving energy usage, with a cost of £82 (t $CO_2e$)$^{-1}$. |
| Anaerobic digestion | Using anaerobic digestion measures could result in up to a 44% reduction in GHG emissions and energy savings of up to 41%, with a saving of £177.30 per tonne of carbon dioxide equivalent for cattle [-£177.3 (t $CO_2e$)$^{-1}$], and £250 for pigs [-£250 (t $CO_2e$)$^{-1}$]. |
| Feed supplements | Supplement feed with feed additives to improve the efficiency of feed utilisation and reduce enteric $CH_4$ emissions and/ or $NH_3$ in their urine. Examples of possible feed additives include lipid and nitrate supplements, extract of liquorice, and oilseed-based preparations. Such measures could result in up to a 45% reduction in $CH_4$ enteric emissions (e.g., for lipid and nitrate dietary supplements), and up to a 77% reduction in $NH_3$ emissions (e.g., for liquorice supplements), whilst improving livestock productivity. There is also a cost of £55.30 per tonne of carbon dioxide equivalent for nitrate [£55.3 (t $CO_2e$)$^{-1}$] and £85.50 for 3NOP [£85.5 (t $CO_2e$)$^{-1}$] |
| Increase fresh grass | Increasing the use of fresh grass for cattle consumption could result in up to 39% reductions in enteric $CH_4$ emissions, whilst improving farm profitability, with no direct costs. |
| Monitor livestock performance | Increasing the extent of monitoring the performance of livestock. For instance, for sheep producers, this measure could reduce carbon footprints by up to 18% whilst increasing economic margins (by £6 per ewe for sheep production). |
| Soil amendments | Applying additional soil amendments (e.g., basalt, biosolids) on the cropland could increase the SOC sequestration by up to 17% for biosolids, and by a 4-fold increase for basalt, whilst increasing crop yield and improving soil fertility, with no direct costs. |
| Nitrogen inhibitors | Applying nitrification inhibitors to the soil can reduce $N_2O$ emissions by 13–53% and decrease nitrate leaching and run-off. It does, however, cost £0.1 (kg N)$^{-1}$ to apply. |
| Introduce legumes | Introducing legumes to grasslands and/or crop rotations could result in up to a 58% reduction in nitrate emissions, with less energy usage for fertilisation, and a saving of £1038 per tonne of carbon dioxide equivalent [-£1038.0 (t $CO_2e$)$^{-1}$]. |
| Cover crops/ minimum tillage | Growing cover crops and switching to reduced/minimum tillage could result in a 25% increase in SOC, whilst enhancing water retention and preventing soil erosion. There is a cost of £124 per tonne of carbon dioxide equivalent [£124 (t $CO_2e$)$^{-1}$]. |
| Trees and hedgerows | Planting more trees and hedgerows on the farm could result in up to 2 times higher SOC concentration, and up to a 53% reduction in $NH_3$ emissions, whilst providing shelter for livestock and helping to conserve soil and water. There is a cost of £55 per tonne of carbon dioxide equivalent [£55 (t $CO_2e$)$^{-1}$]. |
| Land use change | Changing land use such as converting arable land to extensive grassland could result in a 4-fold increase in SOC, whilst also helping to mitigate pollution and nitrate leaching. There is a cost of £170–500 per tonne of carbon dioxide equivalent [£170–500 (t $CO_2e$)$^{-1}$]. |
| Waste for bioelectricity | Using agricultural waste to produce bioelectricity such as litter gasification could potentially provide 0.6% of electricity and heat in the UK and save 1.7 million tonnes of GHG per year. This implies a cost of £34 per tonne of carbon dioxide equivalent [£34 (t $CO_2e$)$^{-1}$]. |
| Bioenergy crops | Growing bioenergy crops (e.g., Willow, Miscanthus) could save up to 53 million tonnes of $CO_2$ by 2050, with a cost of $20–100 per tonne. |

supported with consent forms and participant information sheets underwent review and ethical clearance was awarded by the Countryside and Community Research Institute, University of Gloucestershire ethical research committee.

The survey was advertised publicly in the JISC online survey and was live between 02/12/2022-31/01/2023. Conducting the survey during the winter months was beneficial for maximising participation as it is often the quietest time during the farming calendar. Participants were recruited using a paid-for advertisement in a mainstream agricultural news outlet in the UK (Agricultural Land Managers Weekly), through postings across various social media platforms, the Farming Forum, and Word-of-mouth. Participation was incentivised with a £100 voucher prize draw and written/signed consent was obtained from participants.

One limitation of publicly advertised online surveys with a prize draw incentive is the risk of inaccurate or duplicate responses. The data resulting from the survey was therefore critiqued by the research team, with any entries deemed as duplicates or inconsistent removed from the analysis. This resulted in 56 removals, leaving a remaining sample size of 201. The sample data (n = 201) covered the cereals, dairy, general cropping, grazing livestock (less favoured areas), horticulture, lowland grazing livestock, mixed cropping and specialist pigs and poultry sectors across England, Wales and Scotland. The resulting data were largely quantitative, including some discussions in the optional free-text boxes which were fundamental in understanding the underlying factors driving farmer attitudes towards adoption of GHG mitigation measures.

## 4.2 Estimation strategy

**4.2.1 The dependent variable.** In line with existing studies, and specifically [84,85], the dependent variable was derived as the mean scores of scaled responses across all GHG mitigation measures for each respondent. Farmers' willingness to adopt GHG mitigation measures was described based on a 5-point scale as follows: 1 = very unwilling; 2 = quite unwilling; 3 = neutral, 4 = quite willing and 5 = very willing. This was used to estimate the mean willingness to adopt score for each respondent. For a robust estimate, we further estimated a second dependent variable using Principal Component Analysis (PCA). PCA scores were generated from ranking and the corresponding relative significance of GHG mitigation measures and willingness scales. The PCA scores were estimated using *'pca'* code in STATA. This estimation generates Eigenvalues considering standardised weights of each GHG mitigation measure as follows:

$$|R - \lambda I| W = 0 \qquad (1)$$

where: I represents the identity matrix, $\lambda$ is the vector of the eigenvalues and W is a $p \times p$ dimension matrix containing standardised weight values $w_{ij}$ of each GHG mitigation measure.

A total of 14 eigenvalue components were generated. Table 3 provides descriptive statistics of those principal components with eigenvalues of more than 1 (six components). The coefficient in each component shows the percentage contribution of corresponding variables; for example, the first principal component (PC1) explains a contribution of 14.94% for the corresponding GHG mitigation measures. Figs 1 and 2 illustrate the scree plots showing the mean eigenvalues and the corresponding 95% confidence intervals, respectively. Fig 3 shows a matrix of score variables across the six components. The strength of distribution and commonality of components is illustrated by the degrees of clustering as shown across the matrix. The plotting of the first and second components is, however, considered more important since they have the highest explanatory power.

**Table 3. Descriptive statistics of the principal component analysis estimates.**

| Component | PC1 | PC2 | PC3 | PC4 | PC5 | PC6 |
|---|---|---|---|---|---|---|
| Eigenvalue | 6.063 | 1.514 | 1.061 | 0.858 | 0.731 | 0.688 |
| Difference | 4.549 | 0.452 | 0.204 | 0.127 | 0.043 | 0.068 |
| Proportion | 0.433 | 0.108 | 0.076 | 0.061 | 0.052 | 0.049 |
| Cumulative | 0.443 | 0.541 | 0.617 | 0.678 | 0.731 | 0.780 |

Further, the PCA scores were generated from the weights of variables in each component and standardised variable as follows:

$$s_{kj} = t_{1k}Z_{1j} + t_{2k}Z_{2j} + t_{3k}Z_{3j} + \cdots + t_{pk}Z_{pj} \qquad (2)$$

where $j = 1, 2,\ldots n$, is the number of observations; $k = 1, 2\ldots q$, is the number of selected component numbers; $p$ *is* the number of independent variables; $s_{kj}$ is the standardised score value of $jth$ observations in $kth$ components; $Z_{pj}$ is the standardised value of $pth$ variable of $jth$ observation, calculated from $z = x_p \bar{x}/S_x$, where $x_p$ is the original value of the $pth$ variable, and; $t_{pk}$ the standardised weight of the $pth$ variable in the $kth$ components [86].

Having estimated our dependent variables (Mean willingness score of farmers to adopt and PCA scores (component one) of willingness to adopt GHG mitigation measures), the kernel distribution of both dependent variables is further presented in Figs 4 and 5. The density distribution of the Mean willingness score shows similar skewness to the PCA scores.

To understand the structure of the data collected, we employed descriptive statistics (percentage means and cross-tabulations) (see attached S1 Table), and a test of means (t-test). To assess the factors determining farmers' willingness to adopt GHG mitigation measures, we modelled a multiple regression framework where the dependent variables (mean willingness to adopt score and PCA score) are respectively modelled as a function of farm and farmers' characteristics.

**4.2.2 Analytical framework for determining key factors for farmer willingness to adopt GHG mitigation measures.** The multiple regression analysis was adapted to establish the relationship between farmers' willingness to adopt GHG measures and the explanatory

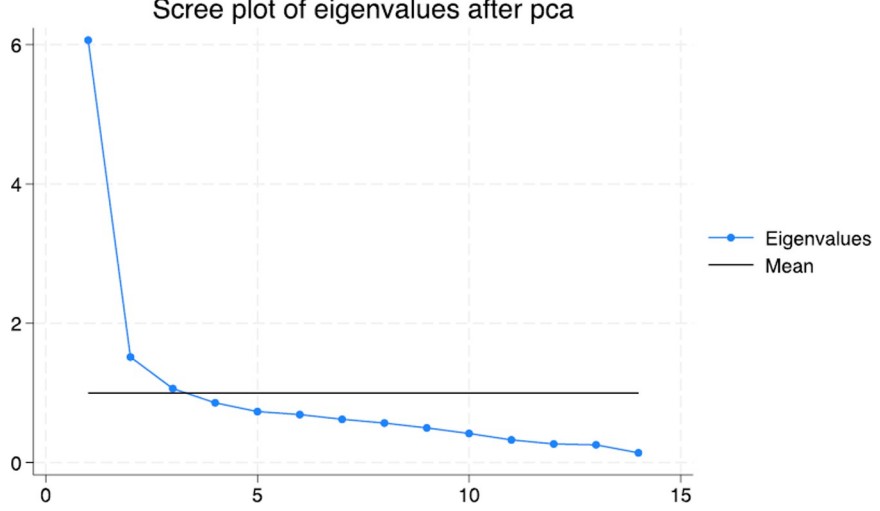

**Fig 1. Scree plot of means and eigenvalues.**

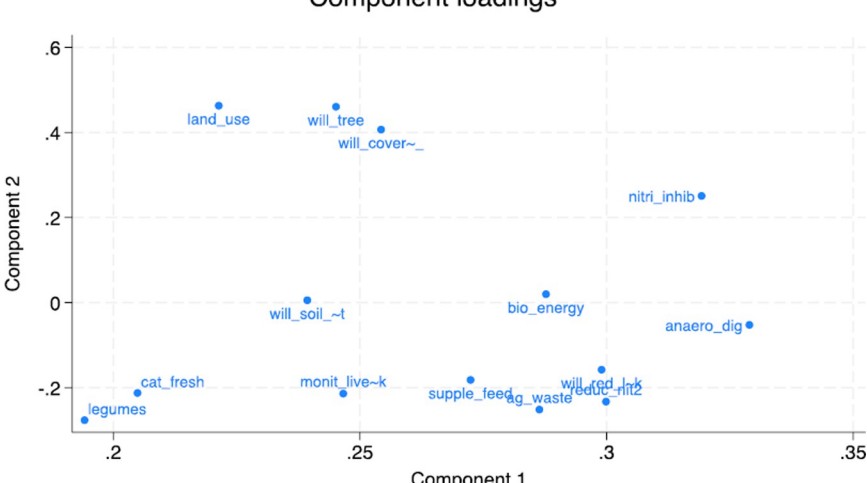

**Fig 2. Illustrations of component loadings.**

variables. The willingness to adopt GHG measures was modelled as a function of explanatory variables and expressed as follows:

$$y_i = \beta_0 + \beta_1 X_1 + \beta_2 X_2 \ldots \beta_n X_n + \varepsilon_n \tag{3}$$

where: $y_i$ respectively represents the mean score of farmers' willingness to adopt GHG measures and PCA scores of farmers' willingness to adopt GHG mitigation measures, $\beta_1 \ldots \beta_n$ represents the vectors of farmers' socioeconomic and farm characteristics ($X_1 \ldots X_n$) to be estimated, $\beta_0$ is the random intercept, and $\varepsilon_n$ is the random error term.

Although linear regression is effective in establishing a relationship between the dependent and independent variables, it is highly susceptible to multicollinearity and high correlation errors. We further tested the significance of estimates in our main regression model by applying stepwise regression models [87,88]. To reduce the bias in model selection in the stepwise regression, we applied the *'vselect'* code in Stata 16 to select the best model [89]. The *'vselect'* -

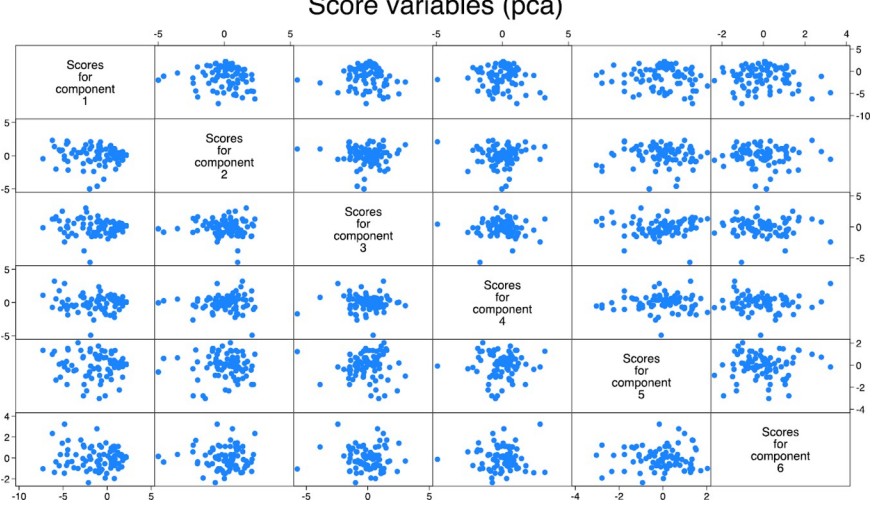

**Fig 3. Matrix of score variables across six components.**

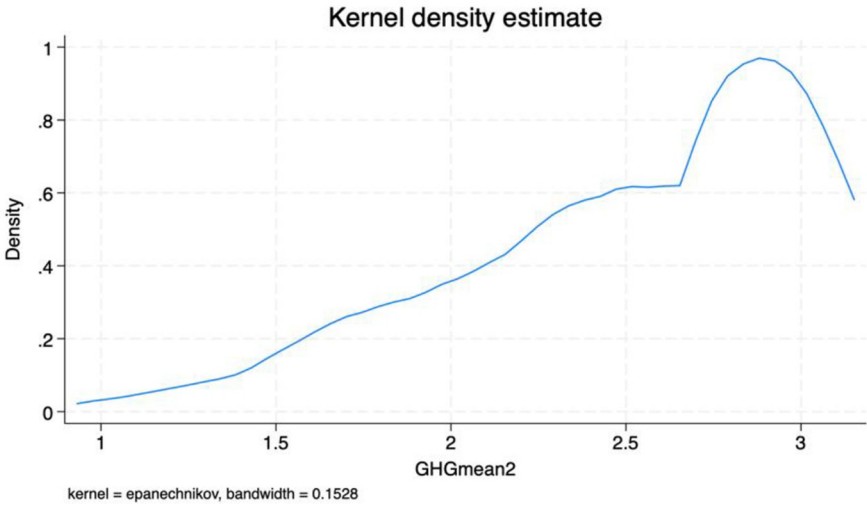

**Fig 4. Kernel density distribution of mean willingness scores.**

performs variable selection for linear regression through the use of the Furnival-Wilson leaps-and-bounds algorithm [90]. The criteria for best model and variable combinations were measured by the significance of the following criterion: the R^2 adjusted, Mallows's C, Akaike's information criterion, Akaike's corrected information criterion, and Bayesian information criterion for the best regression model at each quantity of predictors [90]. We present the estimates of model selection in the attached S2 Table.

## 5.0 Results and discussion

### 5.1 Summary statistics of respondents' socioeconomic characteristics

Table 4 presents the summary statistics of respondents' socio-economic characteristics. On average, 72.6% of respondents were <50 years old and approximately 52% have more than 20 years of farm experience. The sample statistics further show that the agricultural land managers

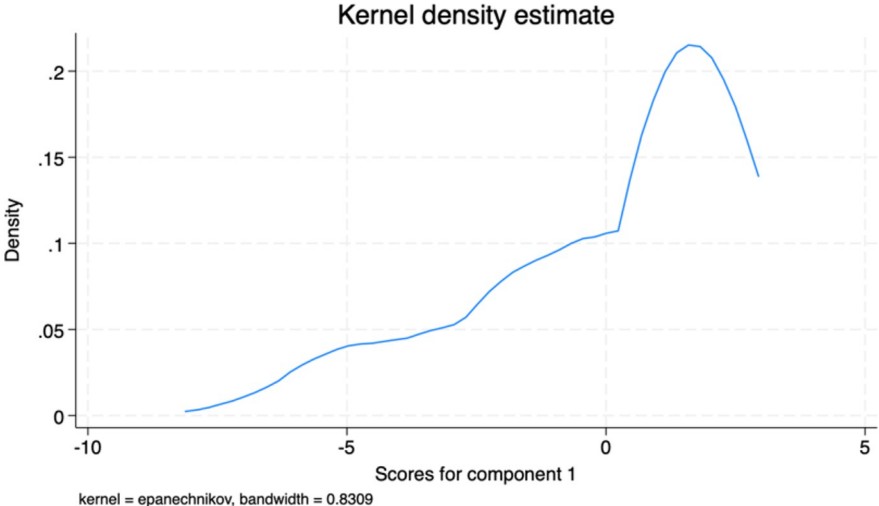

**Fig 5. Kernel density distribution of PCA scores.**

**Table 4. Summary statistics of explanatory variables for farmer attitudes towards adoption of GHG mitigation measures.**

| Variables | Categories | Percentage Distribution | Mean | Std.Dev |
|---|---|---|---|---|
| Age category (below 50 years) | yes = 1 | 72.6 | 0.73 | 0.45 |
| | no = 0 | 27.4 | 0.84 | 0.37 |
| Education | yes = 1 | 84.0 | | |
| | no = 0 | 16.0 | | |
| Farming experience | 0_20 yrs | 47.8 | | |
| | 21 years and above | 52.2 | | |
| Farmed area (ha) | | | 146.80 | 211.87 |
| Rented land(ha) | | | 61.49 | 141.55 |
| Owned land (ha) | | | 87.20 | 115.54 |
| Perception of farm business | well | 57.9 | | |
| | average | 33.3 | | |
| | poorly | 8.8 | | |
| Grow cover crop | yes = 1 | 83.5 | | |
| | no = 0 | 16.5 | | |
| Own livestock | yes = 1 | 89.9 | | |
| | no = 0 | 10.0 | | |
| Adopt Anaerobic digestion | yes = 1 | 34.0 | | |
| | no = 0 | 66.0 | | |
| General cropping | yes = 1 | 15.0 | | |
| | no = 0 | 75.0 | | |
| Grazing livestock | yes = 1 | 14.5 | | |
| | no = 0 | 85.5 | | |
| Mixed farming | yes = 1 | 15.5 | | |
| | no = 0 | 84.5 | | |
| Cereal farming | yes = 1 | 28.0 | | |
| | no = 0 | 72.0 | | |

Source: Authors' computation from survey.

(84%) are highly educated and had varying farm businesses spanning crops and livestock farming constituting 28% cereal farmers, 14% livestock farmers, and 15.5% and 15% mixed and general cropping farmers, respectively. The mean farmed land area for each respondent was 146.80 ha, and the corresponding mean rented and owned land areas were 61.5 ha and 87.2 ha, respectively. The data include some details of existing land use management activities and GHG mitigation measures. For example, approximately 85% grow cover crops and 56% undertake anaerobic digestion, as a manure management practice. While 57.9% of respondents perceived their farm business to be doing quite well, 33.3% placed their farm business in the average category and only 8.8% perceived their farm business to be doing poorly. We further discuss the structure of socioeconomic variables in the attached supporting information (see S1 Table).

## 5.2 Spread of farmers' mean willingness to adopt GHG mitigation measures

Table 5 illustrates the percentage distribution of respondents' willingness to adopt GHG measures collapsed into three categorical measures - 1 = unwilling, 2 = neutral, and 3 = willing. Across all the GHG mitigation measures considered, the descriptive results showed an overall higher percentage of farmers' willingness, to adopt GHG mitigation measures. For example, in

**Table 5. Percentage distribution of farmers' willingness to adopt individual GHG mitigation measures.**

| GHG Mitigation Measures | % distribution of willingness to adopt GHG measures | | |
|---|---|---|---|
| | Willing | Neutral | Un-willing |
| Decrease N fertiliser use | 67.7 | 17.2 | 15.15 |
| Reduce livestock stocking density | 59.4 | 22.4 | 18.2 |
| Anaerobic digestion | 57.0 | 26.2 | 16.7 |
| Supplement feed | 65.7 | 20.8 | 13.5 |
| Increase cattle fresh grass | 74.7 | 21.3 | 4.0 |
| Monitor livestock performance | 68.1 | 23.7 | 8.3 |
| Grow cover crop | 63.7 | 25.2 | 11.1 |
| Soil amendments | 68.9 | 18.1 | 13.0 |
| Nitrogen inhibitors | 60.7 | 25.7 | 13.6 |
| Introduce legumes | 77.5 | 14.5 | 8.1 |
| Tress and hedgerows | 71.6 | 18.3 | 10.2 |
| Land-use change | 59.7 | 26.0 | 14.3 |
| Waste for bioelectricity | 64.1 | 24.6 | 11.3 |
| Bioenergy crops | 58.5 | 24.6 | 16.9 |

Table 4, 56% of respondents are willing to use anaerobic digestate, 25.7% and 16.6% were neutral and unwilling, respectively. In comparison, 68% of respondents were willing to introduce legumes, whereas 12.7% and 7% were neutral and unwilling, respectively.

Table 6 presents estimates of tests of the mean difference of mean willingness scores across farmers' socioeconomic and farm characteristics using the independent sample t-test. The results show significant differences across variables except livestock ownership and farm experience (21–40 years). While the t-test shows possible relationships between farmers' socioeconomic characteristics and willingness to adopt GHG mitigation measures, it does not fully establish causal effects. Causality and determining relationships are further discussed in the next section.

## 5.3 Estimating factors determining farmers' willingness to adopt GHG mitigation measures

Table 7 presents estimates of multiple linear regression and stepwise regression for two outcome models: mean willingness to adopt scores and PCA scores (see method section). The multiple linear regression models show significant model results at $p < 0.01$, suggesting that the explanatory variables explain 46.4% and 50.2% of the variations in farmers' mean willingness to adopt scores and PCA scores, respectively.

Certain predictors were consistently significant across both outcome models. Farm types were, for example, significant determinants of farmers' willingness to adopt certain GHG mitigation measures. To highlight, agricultural land managers whose dominant production was cereal farming were willing to adopt GHG mitigation measures ($p < 0.01$); this result is consistent with agricultural land managers whose dominant agricultural practice is general cropping. In contrast, agricultural land managers whose dominant production is livestock grazing were significantly ($p < 0.01$) unwilling to adopt GHG mitigation measures. An explanation for this difference may be due to the perceived ease of user adoption of GHG reduction measures among cereal and general cropping farmers, compared with livestock production. For livestock farmers, similar findings have highlighted perceived difficulty for the adoption of innovations among livestock farmers in England and Wales [91]. Similarly, [92] reported that livestock farmers' perception of difficulty in investing in animal health technology influenced

**Table 6. Test of mean differences of farmers' socioeconomic characteristics.**

| Variables | Mean difference | t | df | sig |
|---|---|---|---|---|
| Age category (below 50 years) | -0.265*** | -3.509 | 197 | 0.001 |
| Education (category-yes) | 0.520*** | -3.683 | 196 | 0.000 |
| Farmer experience 0–20 years | -0.170** | -2.469 | 198 | 0.014 |
| Farmer experience 21–40 years | 0.077 | 1.067 | 198 | 0.288 |
| Farm perceived well | -0.314*** | -4.730 | 195 | 0.000 |
| Farm perceived average | 0.199*** | 2.770 | 195 | 0.006 |
| Grow cover crop | -0.615*** | -7.219 | 198 | 0.000 |
| Own livestock | 0.011 | 0.097 | 197 | 0.923 |
| Anaerobic | -0.367*** | -5.632 | 198 | 0.000 |
| General cropping | -0.314*** | -3.322 | 198 | 0.001 |
| Grazing livestock | 0.644*** | 7.397 | 198 | 0.000 |
| Mixed farming | 0.198** | 2.083 | 198 | 0.038 |
| Cereals | -0.272*** | -3.615 | 198 | 0.000 |
| Own poultry | -0.316*** | -4.654 | 198 | 0.000 |
| Work(fte) | 0.308*** | 4.568 | 197 | 0.000 |

Source: Authors' computation from survey.

**, *** represents significance at $p < 0.05$, and $p < 0.01$, respectively.

their membership in specific technological uptake groups. Likewise, resource constraint beliefs were found to be significant determinants of dairy farmers' adoption of best management grazing practices in Ireland [93].

Also, agricultural land managers whose farm worker input was less than 2.5fte were less likely to adopt GHG mitigation measures, compared to farm businesses whose worker inputs were >2.5fte; this was significant at *$p < 0.01$*. This result suggests that adopting GHG mitigation measures is highly labour-input dependent. In existing adoption literature, demand for labour is an established barrier to adoption of innovations in agriculture [94–97]. Given the persisting issues of labour shortage in the UK agri-food sector, this result offers insights into the current and long-term effect of labour supply changes on emission reduction policies.

The estimates further show that younger agricultural land managers below 50 years were positively and significantly ($p < 0.05$) more willing to adopt GHG mitigation measures, suggesting that younger agricultural land managers are more flexible and less risk-averse to taking up innovative approaches. A similar study [98] using a representative survey of Scottish farmers, found that younger agricultural land managers were more willing to afforest land. Contrary findings have suggested that the adoption of GHG mitigation measures is more prominent among older agricultural land managers but this was non-significant in some studies [99].

Table 8 presents stepwise regression estimates of the selected best model as described in the method section. In all presented models (except for PCA scores (six predictor model) where age is not significant), estimates are significant and in line with the multiple regression estimates in Table 6 and show the robustness of our regression estimations.

## 5.4. Understanding farmers' willingness to adopt GHG measures–drawbacks and incentives

In addition to the statistical estimation explored above, this section provides insights into supportive explanations provided by agricultural land managers on various mitigation measures.

**Table 7. Multiple regression estimates of willingness to adopt GHG mitigation measures.**

| Predictors | Multiple regression estimates (Mean Willingness Scores) | Multiple regression estimates (PCA Scores) |
|---|---|---|
| | Coef. (Std. Error) | Coef. (Std. Error) |
| Grow cover crop (yes = 1, no = 0) | 0.015 (0.121) | 0.222 (0.917) |
| Age category (below 50 years) (yes = 1, no = 0) | 0.109 (0.085) | 0.295 (0.554) |
| Farmed area (ha) | 2.160E-4 (1.929 E-4) | 3.213E-4 (0.001) |
| Rented land (ha) | 2.221E-4 (2.856E-4) | 7.054E-4 (0.002) |
| Owned land (ha) | 0.001 (0.001) | 0.001 (0.007) |
| Own livestock (yes = 1, no = 0) | -0.054 (0.113) | |
| Anaerobic digestion (yes = 1, no = 0) | 0.199** (0.080) | 0.855* (0.469) |
| Perceived farm is doing well (yes = 1, no = 0) | 0.116 (0.113) | 1.830** (0.776) |
| Perceived farm is average (yes = 1, no = 0) | -0.010 (0.114) | 0.252 (0.798) |
| workers (fte) (0–2.5 fte) | -0.226*** (0.067) | - 1.517*** (0.404) |
| Farm experience category (0–20 years) (yes = 1, no = 0) | 0.104 (0.107) | 0.118 (1.683) |
| Farming experience category (21–40 years) (yes = 1, no = 0) | 0.114 (0.104) | -0.139 (0.705) |
| General cropping (yes = 1, no = 0) | 0.270*** (0.099) | 1.395** (0.541) |
| Cereal farming (yes = 1, no = 0) | 0.244*** (0.080) | 1.570*** (0.470) |
| Own poultry (yes = 1, no = 0) | -0.018 (0.070) | -0.168 (0.436) |
| Livestock grazing (yes = 1, no = 0) | -0.287** (0.118) | -1.799** (0.804) |
| Mixed farming (yes = 1, no = 0) | -0.059 (0.098) | -0.228 (0.619) |
| Education (yes = 1; no = 0) | 0.206 (0.140) | 0.054 (0.828) |
| Constant | 2.072*** (0.261) | -2.137 (1.430) |
| Number of observations | 179 | 122 |
| F Stat | 7.17 | 6.18 |
| Prob > F | 0.000 | 0.000 |
| R-squared | 0.464*** | 0.502*** |
| Adjusted R-squared | 0.404 | 0.421 |

Source: Authors' computation from survey. **, *** *represents significance at, $p < 0.05$, and $p < 0.01$, respectively. Owned livestock estimate in the PCA Score regression model was omitted due to collinearity issues.*

**Table 8. Stepwise regression estimates of willingness to adopt GHG mitigation measures.**

| Predictors | Mean Willingness Score | Mean Willingness Score | PCA Scores | PCA Scores |
|---|---|---|---|---|
| | (7 predictors) Coef. (Std. Error) | (8 predictors) Coef. (Std. Error) | (6 predictors) Coef. (Std. Error) | (7 predictors) Coef. (Std. Error) |
| Grow cover crop (yes = 1, no = 0) | | | | |
| Age category (below 50 years) (yes = 1, no = 0) | 0.148** (0.061) | 0.134** (0.061) | 0.722 (0.462) | 0.794* (0.453) |
| Farmed area (ha) | | | | |
| Rented land (ha) | | | | |
| Owned land (ha) | | | | |
| Own livestock (yes = 1, no = 0) | | | | |
| Anaerobic digestion (yes = 1, no = 0) | 0.211*** (0.056) | 0.193*** (0.056) | 1.238*** (0.442) | 1.021*** (0.441) |
| Perceived farm is doing well (yes = 1, no = 0) | | 0.111 (0.057)* | | |
| Perceived farm is average (yes = 1, no = 0) | | | | |
| workers (fte) (0–2.5 fte) | -0.239*** (0.056) | -0.252*** (0.057) | -1.714*** (0.379) | 1.500*** (0.381) |
| Farm experience category (0–20 years) (yes = 1, no = 0) | | | | |
| Farming experience category (21–40 years) (yes = 1, no = 0) | | | | |
| General cropping (yes = 1, no = 0) | 0.266*** (0.079) | 0.248*** (0.078) | 1.586*** (0.497) | 1.577*** (0.497) |
| Cereal (yes = 1, no = 0) | 0.310*** (0.066) | 0.282*** (0.066) | 2.008*** (0.460) | 1.982*** (0.487) |
| Own poultry (yes = 1, no = 0) | | | | |
| Livestock grazing (yes = 1, no = 0) | -0.288*** (0.086) | -0.272*** (0.087) | -1.880** (0.753) | -1.656** (0.743) |
| Mixed farming (yes = 1, no = 0) | | | | |
| Education (yes = 1; no = 0) | 0.323*** (0.111) | 0.242** (0.117) | | 1.859** (0.738) |
| Constant | 2.046*** (0.133) | 2.100*** (0.136) | -1.347** (0.600) | -3.088*** (0.907) |
| Number of observations | 197 | 195 | 131 | 131 |
| F Stat | 22.13 | 19.77 | 11.89 | 11.54 |
| Prob > F | 0.000 | 0.000 | 0.000 | 0.000 |
| R-squared | 0.450*** | 0.460*** | 0.365*** | 0.396*** |
| Adjusted R-squared | 0.430 | 0.436 | 0.335 | 0.362 |

Source: Authors' computation from survey. **, *and* *** represents significance at, $p < 0.05$, and $p < 0.01$.

We have attached more qualitative insights relating to individual GHG mitigation measures in the S3 Table.

The vital roles of agricultural land managers in protecting the environment and tackling climate change cannot be overemphasised and motivations to adopt GHG mitigation measures vary. From supportive comments, agricultural land managers in most cases are self-aware of the benefits of adopting GHG mitigation measures and this reiterates their responsibilities towards protecting the environment as citizens; a participant explained as follows:

> *Nitrogen fertiliser is our biggest contribution to GHG emissions and water pollution. I also believe that, if we could find alternative forms of crop nutrition, it could improve our crop health and reduce incidence of pests and diseases and our reliance on pesticides*

*(Anonymous respondent)*

*Very open to increasing biodiversity and we are forever gathering acorns, walnuts, pinecones etc and my wife grows the seeds on to whips which we plant. We take local seeds and multiply them up instead of buying in imported species.*

*(Anonymous respondent)*

Also, when asked about their willingness to reduce emissions, some agricultural land managers, for example, articulated their role in cutting down on nitrogen fertilisers to protect the environment and improve water quality through the use of nitrification inhibitors as an important means of reducing farm GHG emissions and ensuring nitrogen use efficiency/reducing pollution. There are also indications of uptake of GHG mitigation measures such as conversion of arable land to pasture, installation of wind turbines, conversion of agricultural waste to bioelectricity, planting many trees and hedgerows and having already introduced mixed legumes into grassland/rotational grazing as explained by one participant:

*I am reducing stocking density as more extreme weather is shortening my grazing season for various reasons. I am experimenting with agroforestry (nut trees) as I think it is important to produce nutrient dense foods in a sustainable way.*

*(Anonymous respondent)*

On the other hand, there are indications that agricultural land managers are still largely in the decision-making phase. We summarise paraphrased responses based on each GHG mitigation measure (see S3 Table). Highlights of motivations and drawbacks to adoption are discussed below:

a. **Cost of adoption and concerns for profitability:** Agricultural land managers identified cost as a limitation to the uptake of certain GHG mitigation measures. For example, in the case of anaerobic digestate, agricultural land managers are of the opinion that anaerobic digestate plants are too expensive for small-scale farmers, and where there is willingness to adopt, transitioning requires investment in lots of infrastructure and with low returns; a respondent explained as follows:

   *We are predominantly a cereal/veg farm, so to move to anaerobic digestate means a considerable investment in infrastructure for which the returns are not there. I would also question the green credentials of AD, is the carbon footprint of the infrastructure manufacturing taken into account anywhere? It is a very energy-intensive operation growing and harvesting the crop and I have serious doubts whether energy produced is greater than total energy consumed.*

   *(Anonymous respondents)*

   Agricultural land managers showed willingness to try out approaches at a domestic level if the opportunity presented itself and if capital grants were available and existing evidence supported the scope for profit. Incentive policies are effective in promoting capital-intensive GHG mitigation measures, e.g. bioelectricity generation, given the refund of capital investment to a farm [100].

*I'm probably not big enough or have enough slurry to make digestion viable in my own right. However, if capital grants were available to build a small-scale digester I'd consider hosting the digester & using slurry from my neighbours.*

*(Anonymous Respondent)*

While cost could be a barrier towards initial investment in GHG mitigation measures, we find that the rising cost of some conventional approaches is driving the switch to GHG mitigation measures; this is especially true for nitrogen fertilisers as explained by some respondents.

*We've already had to reduce due to price of fertiliser. It hasn't been great for us this summer although some of that may be the drought. Not having enough fertiliser means we may not have enough food for our stock and could have to buy in from someone who does have enough Fert. I'm not sure that's a great thing to do. Exploring how much we could reduce without adverse effects would be interesting.*

(Anonymous Respondent)

*We already have reduced N fertiliser use. We built a large slurry lagoon so that we can store more slurry and then make better use of it, thereby reducing the need for N fertiliser. This was a good move ahead of the increase in fertiliser prices! We are not large users of inorganic N at all, but we're always looking to only use what we have* to.

*(Anonymous Respondent)*

**b. Uncertainties, and scepticism:** From agricultural land managers' opinions, uncertainties and scepticism about GHG mitigation measures are driven by a number of factors; this includes poor knowledge of use and understanding of short and long-term impact of adoption of GHG mitigation measures. To highlight, some agricultural land managers stated poor knowledge of the use of anaerobic digesters and were not convinced of the evidence on how they impact GHG emission reductions at the farm level. There were similar opinions about nitrification inhibitors, their use and their effect on soil types. We also found that Agricultural land managers' uncertainties are driven by their beliefs and perceptions about certain GHG mitigation measures. Some agricultural land managers believe that having trees on their farms blocks the sunlight from crops, reduces the fertility of surrounding crops, and affects yield. Also, some agricultural land managers think that GHG mitigation measures are driven towards emission reduction targets with little or no concern about how it impacts profitability. In the case of reducing stocking density, agricultural land managers were of the opinion that although reducing stocking density in grassland management will impact emission reduction, such action may negatively affect their efficiency and profitability and result in poor performance of grassland as a result of undergrazing.

*If I lose one, I reduce my income*

*(Anonymous Respondents)*

*Reducing organic stocking rates would have no effect on decreasing carbon, in fact, it would reduce carbon sequestration in my soil.*

*(Anonymous Respondents)*

Also, there are conceptions about the long-term impact on certain GHG mitigation measures on the environment–application of biosolids for example, there are opinions that they contain microplastics as explained by some participants:

*I don't have cropped land. I wouldn't take biosolids as I have concerns about microplastics.*

*(Anonymous Respondent)*

*Bio-solids are expensive, application can be problematic, and they contain microplastics, not something I would apply to my land. Acquiring something like basalt which is not a local product would cost more in the mining, processing and transport than the benefit it would result in.*

*(Anonymous Respondent)*

Uncertainty about innovations is expected as farming is perceived to be risky, and adoption of a new technology is highly dependent on the farmers, farm and agronomic conditions among others [101]. From farmers' discussions, there is need for on field participatory extension of knowledge and advice about GHG mitigation measures as lack of concern for climate change risk and absence of information are likely barriers to action and/or willingness to adopt GHG mitigation measures [102].

*c.* **Land contracts and market constraints:** Land tenure status is a significant determinant of the adoption of agricultural innovations, especially, long-term soil-improvement technologies [103]. According to agricultural land managers, tenure arrangements are a major drawback to adopting GHG mitigation measures; an example is bioenergy crops. Some agricultural land managers stated that local attempts to adopt bioenergy crops are often unsuccessful due to restrictions in contracts and unwilling landlords.

*I do not think landlord would consent to this.*

*(Anonymous Respondent)*

Bioenergy crops are perennial crops and take longer to harvest, contradicting a need for quick returns on investment. Thus, promoting the adoption of GHG mitigation measures that are long-term may require improvements to land contracts to increase tenant-farmer stability. Motion to kickstart such an approach can be seen in the Environmental Land Management test and trial report for DEFRA which looked at farmer perceptions towards long-term agreements [104].

Besides land tenure drawbacks, agricultural land managers worry about geographically limited markets and costs of transportation in distributing biomass among processing stations. In line with a similar study reported by [57], agricultural land managers raised concerns about poor market availability, especially for bioenergy crops despite their economic and environmental benefits.

## 5.5 Limitations

While this study provides important insights into farmers' perceptions of GHG mitigation measures, there are a few limitations that need to be acknowledged. Firstly, data collected are limited to a point in time with a small sample population which may not have captured the full variation of perception and willingness to adopt GHG measures in the UK. Secondly, this study focused largely on the broad framework of factors of adoption of GHG mitigation

emissions within the different agricultural systems using a pooled outcome of perceptions; however, there are certain variations in farm systems and GHG mitigation measure attributes, including endogenous factors that may impact farmers' adoption decisions; these were not considered. We acknowledge possible weakness in the adopted analytical method; especially the stepwise regression analysis which has potential pitfalls and biased parameter estimation due to inconsistencies in model selection. We tried to minimise this error by testing for the significance of the various models used (see method section and supporting file for details). These current limitations point to the need to extend this research in future studies. Regardless, this current study provides foundational insights into the attitudes towards the current uptake of GHG mitigation measures and the behavioural complexities and barriers to uptake. Our study limitations reiterate the need for further research on different GHG mitigation measures and their adoption constraints. There is the need for long-term assessment of behavioural change in the adoption of GHG mitigation measures using time-varying evaluation techniques to capture the complexities of decision-making in the adoption of innovations.

## 6.0 Conclusion and recommendation

Knowledge and drivers of farmers' willingness to adopt different GHG mitigation measures is still limited in the UK. This study provides foundational insights into barriers towards adopting GHG mitigation measures in the agricultural sector, and one of the key highlights of this study is the distinct variations in adoption willingness across crop and livestock sectors, suggesting that policies may have to consider heterogeneities of farm sector needs to deliver rapid adoption of GHG mitigation measures. This study highlighted problems of limiting farm resource factors such as labour. Shortage of agricultural labour is an ongoing challenge in the UK, post departure from the European Union. Our research provides further evidence of the need for long-term impact evaluation of labour shortages on meeting reducing GHG emissions in the agricultural sector and meeting net zero policies. Our research evidenced the need for policy amendments to allow flexible land contracts, as this is central to the uptake of certain GHG mitigation measures, e.g., agroforestry practices and bioenergy crops. This land tenancy factor also extends to on-farm cost-intensive approaches, e.g., the use of anaerobic digesters and long-term soil and organic amendments that require long-term land contracts and flexibility in land use. As a result, a long-term policy promoting incentives may be more beneficial for sustainable uptake of GHG mitigating measures.

This research further underscored limitations associated with poor awareness and knowledge of GHG mitigation measures, including their impact on business-as-usual farming operations. Here, we found evidence of polarisation in knowledge and adoption of GHG mitigation measures, with some agricultural land managers indicating 100% awareness and transition in land use and adoption of GHG mitigation measures, and some respondents stating very limited awareness. There is a need for a mechanism to promote farm-level awareness and participation or on-farm demonstration of each GHG mitigation measure to improve willingness to adopt among farmers. In addition, cost of uptake is, in most cases, a critical barrier, and uncertainty about the outcomes of GHG mitigation measures prevails. Policy can help in varying ways; this can include support for on-farm experimentation by farmers to ease off initial setup and to facilitate adoption. In addition, a landscape model to facilitate co-adoption is important and further research is needed to understand and design mechanisms that can facilitate co-adoption.

## Supporting information

**S1 Fig. Figure of the frequency distribution of willingness to try new measures for reducing GHG emissions.**
(DOCX)

**S1 Table. Frequency distribution of agricultural land managers' socioeconomic characteristics.**
(DOCX)

**S2 Table. Test for stepwise regression predictors selection.**
(DOCX)

**S3 Table. Quotes and paraphrasing of farmers' explanation of willingness to adopt GHG mitigation measures.**
(DOCX)

## Author Contributions

**Conceptualization:** Asma Jebari.

**Data curation:** Charlotte-Anne Chivers.

**Formal analysis:** Zainab Oyetunde-Usman.

**Funding acquisition:** Asma Jebari, Graham A. McAuliffe, Adrian L. Collins.

**Investigation:** Asma Jebari, Graham A. McAuliffe, Adrian L. Collins.

**Methodology:** Zainab Oyetunde-Usman.

**Resources:** Graham A. McAuliffe, Charlotte-Anne Chivers.

**Software:** Zainab Oyetunde-Usman.

**Supervision:** Graham A. McAuliffe, Adrian L. Collins.

**Visualization:** Asma Jebari.

**Writing – original draft:** Asma Jebari, Zainab Oyetunde-Usman.

**Writing – review & editing:** Zainab Oyetunde-Usman, Charlotte-Anne Chivers, Adrian L. Collins.

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
