## [Decision Letter · Decision Letter 0]

26 Feb 2024

PONE-D-23-40470Examining agricultural land manager willingness to adopt climate change mitigation measures in the UK.PLOS ONE

Dear Dr. Oyetunde-Usman,

Thank you for submitting your manuscript to PLOS ONE. After careful consideration, we feel that it has merit but does not fully meet PLOS ONE’s publication criteria as it currently stands. Therefore, we invite you to submit a revised version of the manuscript that addresses the points raised during the review process.

We look forward to receiving your revised manuscript.

Kind regards,

Muhammad Aamer Mehmood, Ph.D.

Academic Editor

PLOS ONE

Journal Requirements:

The first Author - AJ, was financially supported by the Science Initiative Catalyst Award (SICA) programme, a Rothamsted Research internal UKRI (UK Research and Innovation) Biotechnology and Biological Sciences Research Council (UKRI-BBSRC) award. GM and ALC acknowledge funding from Rothamsted Research’s Institutional Strategic Programmes ‘Soil to Nutrition’ (S2N) supported by UKRI-BBSRC BBS/E/C/000I0320 & BBS/E/C/000I0330 and ‘Resilient Farming Futures’ supported by UKRI-BBSRC BB/X010961/1.

AJ was financially supported by the Science Initiative Catalyst Award (SICA) programme, a Rothamsted Research internal UKRI (UK Research and Innovation) Biotechnology and Biological Sciences Research Council (UKRI-BBSRC) award. GM and ALC acknowledge funding from Rothamsted Research’s Institutional Strategic Programmes ‘Soil to Nutrition’ (S2N) supported by UKRI-BBSRC BBS/E/C/000I0320 & BBS/E/C/000I0330 and ‘Resilient Farming Futures’ supported by UKRI-BBSRC BB/X010961/1.

The first Author - AJ, was financially supported by the Science Initiative Catalyst Award (SICA) programme, a Rothamsted Research internal UKRI (UK Research and Innovation) Biotechnology and Biological Sciences Research Council (UKRI-BBSRC) award. GM and ALC acknowledge funding from Rothamsted Research’s Institutional Strategic Programmes ‘Soil to Nutrition’ (S2N) supported by UKRI-BBSRC BBS/E/C/000I0320 & BBS/E/C/000I0330 and ‘Resilient Farming Futures’ supported by UKRI-BBSRC BB/X010961/1.

6. In the online submission form, you indicated that Data access will be provided upon request.

Reviewers' comments:

Reviewer's Responses to Questions

**Comments to the Author**

1. Is the manuscript technically sound, and do the data support the conclusions?

Reviewer #1: Yes

Reviewer #2: Yes

2. Has the statistical analysis been performed appropriately and rigorously? 

Reviewer #1: Yes

Reviewer #2: Yes

3. Have the authors made all data underlying the findings in their manuscript fully available?

Reviewer #1: No

Reviewer #2: Yes

4. Is the manuscript presented in an intelligible fashion and written in standard English?

Reviewer #1: Yes

Reviewer #2: Yes

5. Review Comments to the Author

Reviewer #1: Congratulations to the authors for a well designed and timely piece of research, which contains many interesting findings which I believe will be of interest to both the academic and agricultural communities.

I have a few comments which I believe should be addressed before this research is ready for publication.

Firstly, I think there needs to be much more consideration of the limitations of the chosen analysis here. The authors are clearly aware of the debate over the use of means based on likert-style (ordinal) data. While I do not have a problem with how this has been done here, I have low confidence in the results because there is very limited information on the data distribution, spread etc. There is also no discussion of the ordinal nature of the data, whereby the scores are not equidistant making grouping, calculation of means and use in models dangerous - all of which have been conducted here without allowing for proper scrutiny. Some clearer, simpler information on frequency distributions of the underlying data would be welcome here.

Furthermore, I am slightly uncomfortable at the subsequent use of stepwise regression, which has potential pitfalls including biased parameter estimation, model selection inconsistencies, and inappropriate focussing on one 'best' model or even multiple hypothesis testing. I would like to see more discussion, or even further testing of the degree to which these have occurred and therefore how trustworthy the results are.

Finally, I cannot see a copy of the questionnaire that was delivered, which is important to be able to scrutinise. Especially since some of the responses given in the supplementary material were questioning the posing of some questions. This should be provided, as should more clarity on the underlying data behind the means and percentages given (see comment above here too).

Please see below for some additional comments relating to the line numbers given. The manuscript would benefit from further proof reading to pick up on a few typos.

Line 10: Initial should be AJ?

44: delete "were needed"

122: This sentence mixes up a range of causes and effects and does not make a good link between grazing intensity, biodiversity or pollination services. Suggest the authors make a broader statement, backed up with a recent review of the effects of livestock grazing on biodiversity including pollination (e.g. Schils et al. 2022, Agr. Eco. Env., 330; 107891)

133: Not clear what this % refers to. Potential of fertilisation reductions to reduce total GHG emissions? Suggest clarify.

134: delete comma after reference

177: cover (not covers)

183: change comma to full stop

192-205: This paragraph is lacking some balanced evaluation of the impacts of bioenergy crop production. Whilst Overall this may be true (e.g. https://iopscience.iop.org/article/10.1088/1748-9326/ac22be/meta) it is context dependent and dependent on previous land use (e.g. https://onlinelibrary.wiley.com/doi/full/10.1111/gcbb.12067).

236: I would like to see more detail of these removals. How many? What were some of the examples removed?

243: It is not clear to me where some of these figures come from, and therefore cannot be well scrutinised. E.g. where is the 70.6% figure from and what does it specifically refer to? What is it comparing? SImilarly - where does the 39% figure and no direct costs come from against increasing fresh grass?

331: As above it is currently unclear how some of these figures have been derived. For example, the figures next to "own livestock" seem different from those mentioned in the text. Please clarify or outline the discrepancy.

370: "whose"

382-387: The effect of labour availability is an interesting, if not surprising result. It would be nice to see this set in the current context of labour supply changes following the UK's exit from the EU and other longer term changes to labour supply. Suggest an expansion of discussion here.

Reviewer #2: This article provides an interesting survey on adoption of GHGs mitigation policies by farmers in UK. Overall, it seems considerable for publication in PLOS ONE, however, before it is considered for publication, it needs to be revised and improved. There are several other studies published on this topic, what is the rationale for the current study? Did you find some different results? Could you provide a comparison with other studies?

1. Revise the title to make it attractive.

2. There are a lot of grammatical and syntax mistakes throughout the manuscripts, please revise carefully. Rewrite sentences: L137, L150, L154-156, L168-172 (break into smaller sentences), 206-207,

3. Write objectives of the study in the last paragraph of introduction.

4. Flow of discussion is missing, try to cite more relevant literature and discuss properly.

5. Section headings are not meaningful and attractive, e.g. the first heading of Methods is "Data" , which does not convey the message properly. Authors are advised to revise the sections headings carefully.

6. Revise the conclusion and recommendation section, authors are encouraged to give their viewpoints to cover the gap they mentioned in that section.

6. PLOS authors have the option to publish the peer review history of their article (what does this mean?). If published, this will include your full peer review and any attached files.

Reviewer #1: No

Reviewer #2: No

---

## [Author Response · Author response to Decision Letter 0]

31 May 2024

Reviewer One 

Reviewer #1:Congratulations to the authors for a well-designed and timely piece of research, which contains many interesting findings which I believe will be of interest to both the academic and the agricultural communities. 

I have a few comments which I believe should be addressed before this research is ready for publication. 

1. Firstly, I think there needs to be much more consideration of the limitations of the chosen analysis here. The authors are clearly aware of the debate over the use of means based on likert-style (ordinal) data. While I do not have a problem with how this has been done here, I have low confidence in the results because there is very limited information on the data distribution, spread etc. There is also no discussion of the ordinal nature of the data, whereby the scores are not equidistant making grouping, calculation of means and use in models dangerous - all of which have been conducted here without allowing for proper scrutiny. Some clearer, simpler information on frequency distributions of the underlying data would be welcome here. Furthermore, I am slightly uncomfortable at the subsequent use of stepwise regression, which has potential pitfalls including biased parameter estimation, model selection inconsistencies, and inappropriate focussing on one 'best' model or even multiple hypothesis testing. I would like to see more discussion, or even further testing of the degree to which these have occurred and therefore how trustworthy the results are. 

Response: Thank you very much for your comment, we have included discussion on limitation of our model see line 509 - 524. Also, we have in addition provided the distribution of some data in the attached supplementary file (see Table S1-S7) – which shows the percentage spread of the distribution of GHG mitigation measures.to show the spread of variables used in these studies. 

To the second question, we agree with the inconsistency regarding stepwise regression. Our use of stepwise regression in this context is to further provide useful insights into the relationship between our outcome variables and predictors for comparison with our multiple regression estimates (we have stated this in the method). To ascertain the right combination of variables for the stepwise regression, we have adapted a model selection approach using ‘vselect’ code in Stata 16 to show the varying model combinations and the significant best model, this result is presented in the supplementary file, Table S11- S12 (see shaded yellow column showing the R2ADJ, C, AIC, AICC and BIC model for both outcome models). 

2. Finally, I cannot see a copy of the questionnaire that was delivered, which is important to be able to scrutinise. Especially since some of the responses given in the supplementary material were questioning the posing of some questions. This should be provided, as should more clarity on the underlying data behind the means and percentages given (see comment above here too). 

Response: Thank you very much for your comment. We have attached a copy of questionnaire to our resubmission as requested. 

Please see below for some additional comments relating to the line numbers given. The manuscript would benefit from further proofreading to pick up on a few typos. 

3. Line 10: Initial should be AJ? 

Response: Yes, the initial is AJ – for the first author Asma Jebari , this is now corrected. See line 11. 

4. Line 44: delete "were needed" 

Response: Thank you for your comment, we have reviewed this paragraph in the abstract section. 

5. Line 122: This sentence mixes up a range of causes and effects and does not make a good link between grazing intensity, biodiversity or pollination services. Suggest the authors make a broader statement, backed up with a recent review of the effects of livestock grazing on biodiversity including pollination (e.g. Schils et al. 2022, Agr. Eco. Env., 330; 107891) 

Response: Thank you for your comment, we have carefully reviewed this paragraph, please see paragraph o (line 100 to 111). 

6. Line 133: Not clear what this % refers to. Potential of fertilisation reductions to reduce total GHG emissions? Suggest clarify. 

Response: Thank you very much for your comment, this paragragh has been reviewed. Please see line 108 to 111 

7. Line 134: delete comma after reference 

Response: Thank you for your comment. We have reviewed this line, see line 108 – 110 

8. Line 177: cover (not covers) 

Response: Thank you for your comment, we have now corrected this error – ‘Planting cover crop’ see paragraph - Please see paragraph line 144- 150. 

9. Line 183: change comma to full stop 

Response: Thank you for your comment. Now reviewed. Please see sentence in line 147 – 150. 

10. Line 192-205: This paragraph is lacking some balanced evaluation of the impacts of bioenergy crop production. Whilst Overall this may be true (e.g.https://iopscience.iop.org/article/10.1088/1748-9326/ac22be/meta) it is context dependent and dependent on previous land use (e.g.https://onlinelibrary.wiley.com/doi/full/10.1111/gcbb.12067). 

Response: Thank you very much for your comment. This has been included in the paragraphs, please see line 165-172 - the paragraph on bioenergy crops. 

11. Line 236: I would like to see more detail of these removals. How many? What were some of the examples removed? 

Response: The number of removals are 56 and were duplicated responses, largely missing responses this was done while collating all responses and were not included. 

12. Line 243: It is not clear to me where some of these figures come from, and therefore cannot be well scrutinised. E.g. where is the 70.6% figure from and what does it specifically refer to? What is it comparing? Similarly - where does the 39% figure and no direct costs come from against increasing fresh grass? 

Responses: Thank you for your response. Table 2 was part of the description for our surveys and were all adapted from Jebari et al. 2024. We have cited this in the document. see Table 2. 

13. Line 331: As above it is currently unclear how some of these figures have been derived. For example, the figures next to "own livestock" seem different from those mentioned in the text. Please clarify or outline the discrepancy. 

Response: Thank you very much for your response. Table 3 is a summary statistics of explanatory variables used in this study, we have included a column describing each variable. See line 241 Table Table 2. Also, we have attached in the supplementary data, description/spread of some of the data ( see Table (S1- S8). 

14. Line 370: "whose" 

Response: Thank you for your comment, we have now corrected this. See line 353 -355. 

15. Line 382-387: The effect of labour availability is an interesting, if not surprising result. It would be nice to see this set in the current context of labour supply changes following the UK's exit from the EU and other longer-term changes to labour supply. Suggest an expansion of discussion here. 

Response: Thank you very much for your comment. We have included this statement. Please see the paragraph line 365-370 

Reviewer Two 

16. This article provides an interesting survey on adoption of GHGs mitigation policies by farmers in UK. Overall, it seems considerable for publication in PLOS ONE, however, before it is considered for publication, it needs to be revised and improved. There are several other studies published on this topic, what is the rationale for the current study? Did you find some different results? Could you provide a comparison with other studies? 

Response: Thank you very much for your comments, we have included a section on summary of past studies to highlight findings. See Table 1 and section 3.0 line 179 – 205 

17. Revise the title to make it attractive. 

Response: Thank you for your comment, we have changed this to ‘ Willingness to Adopt Climate Change Mitigation Measures in the UK: A case study of agricultural land managers in the United Kingdom. 

18. There are a lot of grammatical and syntax mistakes throughout the manuscripts, please revise carefully. Rewrite sentences: L137, L150, L154-156, L168-172 (break into smaller sentences), 206-207, 

Response: Thank you very much for your comment, we have addressed all errors, please see the following for reference: 

L137 - we have reviewed this paragraph all together, please (see line 107-110). 

L150 – We have reviewed this paragraph (see line 111 – 122) 

 L154-156 – This paragraph has been reviewed (see line 153 – 163). 

 L168-172 (break into smaller sentences) – The soil amnedments paragragh now reviewed – please see line 166 – 170. 

L206-207 – We have removed this paragraph as it is a repetition of anaerobic digestion discussed under enteric fermentation , please see paragraph 111- 122. 

19. Write objectives of the study in the last paragraph of introduction. 

Response: Thank you very much for your comment, we have indicated this in the last paragraph of the introduction. See line 86 - 88 

20. Flow of discussion is missing, try to cite more relevant literature and discuss properly. 

Response: Thank you very much, we have reviewed the result and discussion section 

21. Section headings are not meaningful and attractive, e.g. the first heading of Methods is "Data" , which does not convey the message properly. Authors are advised to revise the section headings carefully. 

Response: Thank you for your comments. All section head title has been revised accordingly 

22. Revise the conclusion and recommendation section, authors are encouraged to give their viewpoints to cover the gap they mentioned in that section. 

Response: Thank you very much for your comment, we have now thoroughly reviewed thre conclusion and recommendation section

---

## [Decision Letter · Decision Letter 1]

19 Jun 2024

Willingness to AdoptGreen House GasMitigation Measures: Agricultural land managers in the United Kingdom

PONE-D-23-40470R1

Dear Dr. Oyetunde-Usman,

We’re pleased to inform you that your manuscript has been judged scientifically suitable for publication and will be formally accepted for publication once it meets all outstanding technical requirements.

Kind regards,

Muhammad Aamer Mehmood, Ph.D.

Academic Editor

PLOS ONE

Reviewers' comments:

Reviewer's Responses to Questions

**Comments to the Author**

1. If the authors have adequately addressed your comments raised in a previous round of review and you feel that this manuscript is now acceptable for publication, you may indicate that here to bypass the “Comments to the Author” section, enter your conflict of interest statement in the “Confidential to Editor” section, and submit your "Accept" recommendation.

Reviewer #1: All comments have been addressed

Reviewer #2: All comments have been addressed

2. Is the manuscript technically sound, and do the data support the conclusions?

Reviewer #1: Yes

Reviewer #2: Yes

3. Has the statistical analysis been performed appropriately and rigorously? 

Reviewer #1: Yes

Reviewer #2: Yes

4. Have the authors made all data underlying the findings in their manuscript fully available?

Reviewer #1: Yes

Reviewer #2: Yes

5. Is the manuscript presented in an intelligible fashion and written in standard English?

Reviewer #1: Yes

Reviewer #2: Yes

6. Review Comments to the Author

Reviewer #1: Thank you to the authors for their thorough and detailed responses to my questions, which have been addressed well with the amendments to the manuscript. The caveats within the discussion and the extra information in the supplementary material have allowed for far more scrutiny of the underlying data.

Reviewer #2: (No Response)

7. PLOS authors have the option to publish the peer review history of their article (what does this mean?). If published, this will include your full peer review and any attached files.

Reviewer #1: **Yes: **Richard Francksen

Reviewer #2: No

---

## [Editor Report · Acceptance letter]

28 Jun 2024

PONE-D-23-40470R1 

PLOS ONE

Dear Dr. Oyetunde-Usman, 

I'm pleased to inform you that your manuscript has been deemed suitable for publication in PLOS ONE. Congratulations! Your manuscript is now being handed over to our production team.

Kind regards, 

on behalf of

Prof. Muhammad Aamer Mehmood 

Academic Editor

PLOS ONE